# KnobGen: Controlling the Sophistication of Artwork in Sketch-Based Diffusion Models

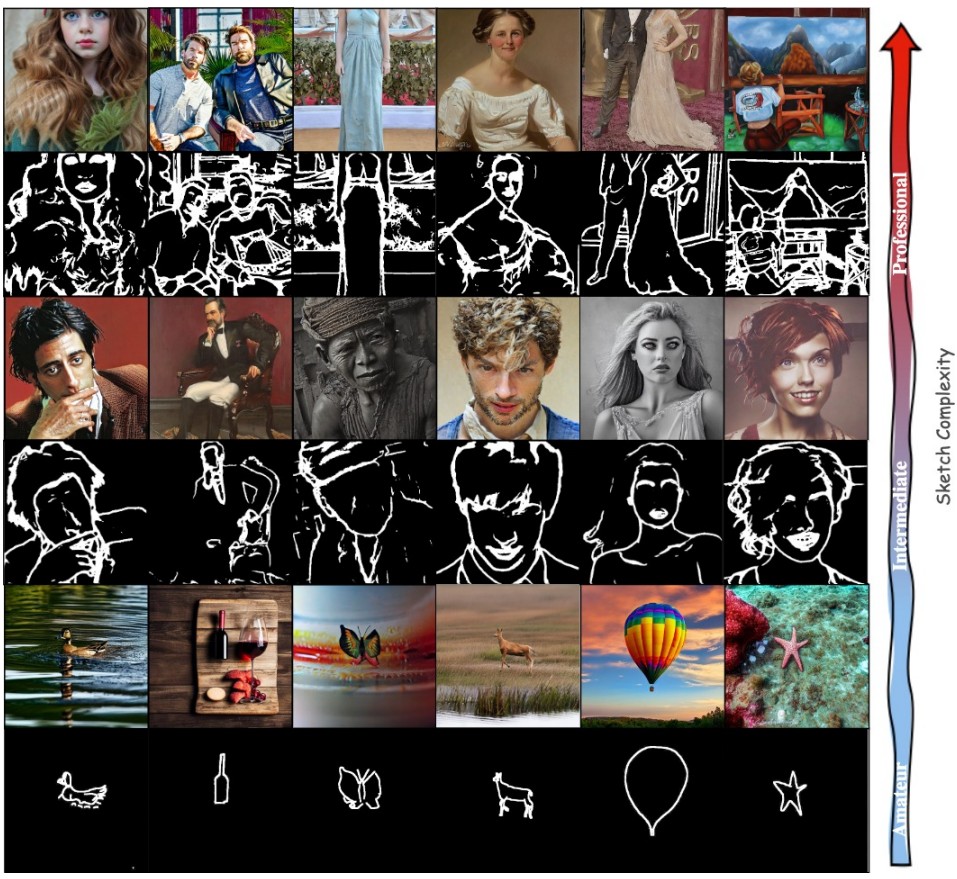

Figure 1: **KnobGen**. Our method democratizes sketch-based image generation by effectively handling a broad spectrum of sketch complexity and user drawing ability—from novice sketches to those made by seasoned artists—while maintaining the natural appearance of the image.

## Abstract

Recent advances in diffusion models have significantly improved text-to-image (T2I) generation, but they often struggle to balance fine-grained precision with high-level control. Methods like ControlNet and T2I-Adapter excel at following sketches by seasoned artists but tend to be overly rigid, replicating unintentional flaws in sketches from novice users. Meanwhile, coarse-grained methods, such as sketch-based abstraction frameworks, offer more accessible input handling but lack the precise control needed for detailed, professional use. To address these limitations, we propose **KnobGen**, a dual-pathway framework that democratizes sketch-based image generation by seamlessly adapting to varying levels of sketch complexity and user skill. KnobGen uses a *Coarse-Grained Controller* (CGC) module for high-level semantics and a *Fine-Grained Controller* (FGC) module for detailed refinement. The relative strength of these two modules can be adjusted

through our **knob** inference mechanism to align with the user's specific needs. These mechanisms ensure that KnobGen can flexibly generate images from both novice sketches and those drawn by seasoned artists. This maintains control over the final output while preserving the natural appearance of the image, as evidenced on the MultiGen-20M dataset and a newly collected sketch dataset.

# 1    INTRODUCTION

Diffusion models (DMs) have revolutionized text-to-image (T2I) generation by generating visually rich images based on text prompts, excelling at capturing various levels of detail—from textures to high-level semantics (Saharia et al., 2022; Rombach et al., 2022; Nichol et al., 2021; Ramesh et al., 2021; Navard & Yilmaz, 2024). Despite their success, one of the primary limitations of these models is their inability to precisely convey spatial layout of the user-provided sketches. While text prompts can describe scenes, they struggle to capture complex spatial features, which makes it challenging to align generated images with user intent. This is particularly intensified when these users vary in skill and experience (Chowdhury et al., 2023; Song et al., 2017; Yang et al., 2023; Jiang et al., 2024).

To improve spatial control, sketch-conditioned DMs like ControlNet (Zhang et al., 2023), T2I-Adapter (Mou et al., 2024), and ControlNet++ (Li et al., 2024) have introduced mechanisms to allow users to input sketches that guide the generated image. However, these approaches primarily cater to artistic sketches with intricate details, which poses a challenge for novice users. When presented with rough sketches, these models rigidly align to unintentional flaws, producing results that misinterpret the user's intent and fail to achieve the desired visual outcome. Furthermore, we observed that the quality and alignment of the generated images with the input sketch are highly sensitive to the weighting parameter that governs the model's dependence on the condition, Figure 2.

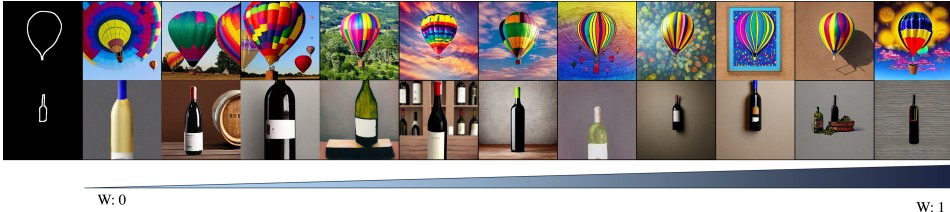

Figure 2: **Qualitative results demonstrating the impact of varying the weighting scheme in T2I-Adapter model**. Lower weights result in images that poorly align with the input sketch in terms of spatial conformity, while higher weights improve spatial conformity of the generated image to the input sketch. However, higher weight compromises the natural appearance of the generated images.

In contrast, some frameworks like (Koley et al., 2024)[1] have attempted to address the needs of novice users by introducing sketch abstraction. Although this democratizes the generation process, Koley et al. (2024) is limited to covering only 125 categories of sketch subjects and cannot handle unseen categories, significantly limiting the generalizability of the pre-trained DM to a limited number of subjects. Moreover, its abstraction-aware framework is not suitable for artistic-level sketches whose purpose is to guide the DM to follow a particular spatial layout. Additionally, the removal of the text-based conditioning in DM makes these models ignore the semantic power provided by text in diffusion models trained on large-scale image-text pairs. Additionally, it limits their ability to differentiate between visually similar but semantically distinct objects- such as zebra and horse.

In a nutshell, existing methods for sketch-based image generation tend to focus on either end of the user-level spectrum. Professional-oriented models like ControlNet and T2I-Adapter are designed to handle only artistic-grade sketches Fig. 3.a, while amateur-oriented approaches Koley et al. (2024), cater to novice sketches without text guidance Fig. 3.b. These methods often fail to integrate both fine-grained and coarse-grained control, limiting their adaptability across different user types and sketch complexities.

To address these challenges, we propose **KnobGen**, a dual-pathway framework designed to empower a pre-trained DM with the capability to handle both professional and amateur-oriented ap-

---

[1]The code and model weights at the time of submission were unavailable, preventing result reproduction.

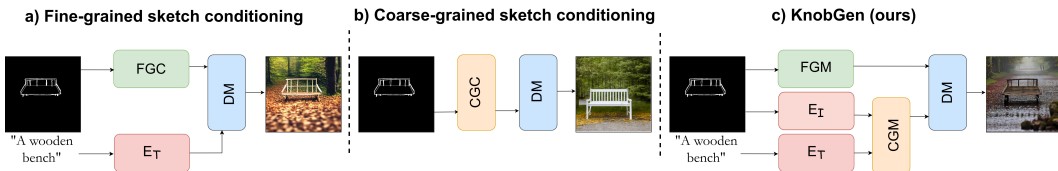

Figure 3: **Comparison across various sketch-control in DM.** (a) fine-grained control based method such as ControlNet or T2I-adapter rigidly resembles a novice sketch resulting in an unrealistic image (b) abstraction-aware frameworks such as Koley et al. (2024) fails to capture fine grained-detials without text guidance(c) while our proposed KnobGen smoothes out the imperfection of the user drawing and preserves the features of the novice sketch. FGC: Fine-grained Controller, CGC: Coarse-grained Controller, $E_T$: Text Encoder, $E_I$: Image Encoder, DM: Diffusion Model.

proaches. KnobGen seamlessly integrates fine-grained and coarse-grained sketch control into a unified architecture, allowing it to adapt to varying levels of sketch complexity and user expertise. Our model is built on two key pathways, *Macro Pathway* and *Micro Pathway*. The Macro Pathway extracts the high-level visual and language semantics from the sketch image and the text prompt using CLIP encoders and injects them into the DM via our proposed **Coarse-Grained Controller** (CGC). The Micro Pathway injects low-level, detailed features and semantics directly from sketch through our **Fine-Grained Controller** (FGC) module.

Additionally, we propose two new approaches for training and inference in order to maintain a robust control of the Micro and Macro Pathways in the conditional generation. First, we introduce **Modulator**, a mechanism dynamically adjusting the influence of the FGC during training, ensuring that the CGC dominates in the early training phase to prevent overfitting to low-level sketch features extracted by the FGC module. This allows the model to optimally rely on both Pathways to capture high- and low-level spatial and semantic features. At inference, the **Knob** mechanism offers user-driven control during denoising steps, allowing adjustment of the level of fidelity between the generated image and the user's inputs- sketch and text- by manipulating Micro and Macro Pathways. These new training and inference approaches ensure that KnobGen effectively handles not only novice sketches but also artistic-grade ones, adapting to varying levels of sketch complexity and user preferences.

Our key contributions are as follows:

- **Dual-Pathway Framework for Sketch-Based Image Generation**: KnobGen proposes an add-on framework to DMs that handles image generation flexibly for a broad spectrum of users, from novice to seasoned artists, through our proposed micro-macro path.

- **Training Modulator for Balanced Coarse-Fine Grained Incorporation**: Our dynamic modulator regulates the influence of our CGC and FGC modules throughout the training. This prevents premature overfitting to fine details, allowing the model to first capture high-level spatial coherence before gradually introducing more specific features.

- **Inference Knob for Adaptive Image Generation**: Our inference-time Knob mechanism allows users to control the level of fidelity between the generated image and the inputs, i.e. sketch image and text. Our Knob mechanism tweaks the level of abstraction and details of the generated image adaptively based on the user's preference.

## 2 RELATED WORK

### 2.1 DIFFUSION MODELS

Recent advances in DM have enabled high-quality image generation with improved sample diversity (Ho et al., 2020; Dhariwal & Nichol, 2021; Ho & Salimans, 2021; Nichol et al., 2021; Saharia et al., 2022; Shamshiri et al., 2024; Ramesh et al., 2022; Perera et al., 2024; Peebles & Xie, 2023), often exceeding the performance of Generative Adversarial Networks (GAN) (Goodfellow et al., 2014; Karras et al., 2019; 2021; Sauer et al., 2022). DMs are built on the concept of diffusion processes, where data are progressively corrupted by noise over several timesteps. The models learn to reverse

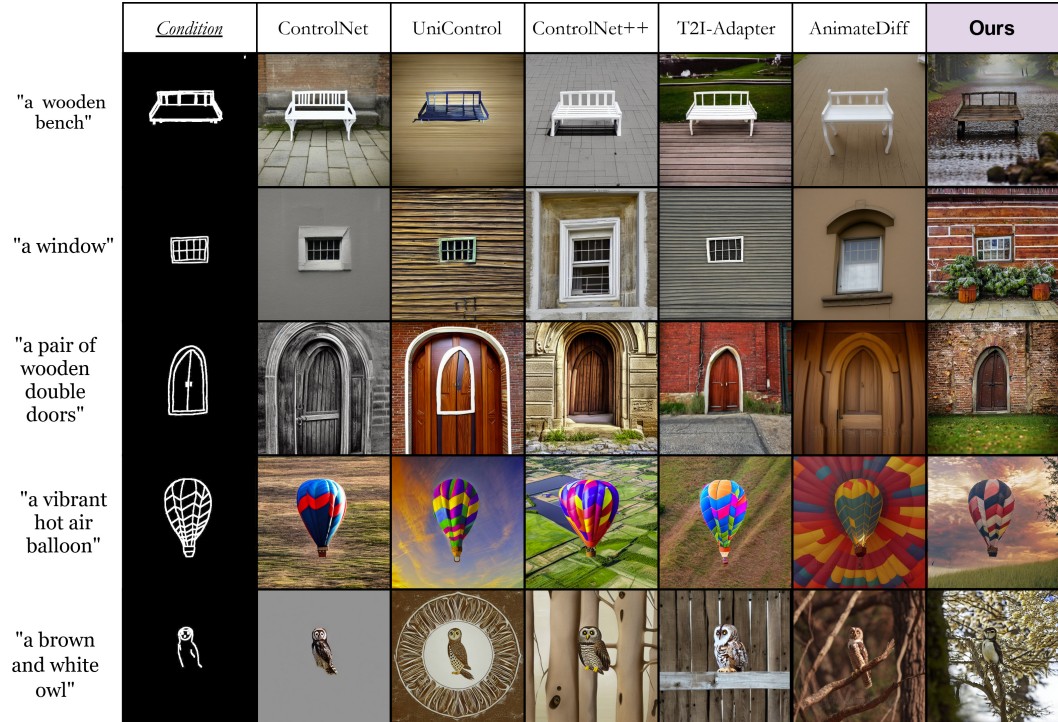

Figure 4: **KnobGen vs. baseline on novice sketches**. KnobGen handles amateurish sketches by injecting features from the Micro and Macro Pathways in a controlled manner. Dual pathway design ensures that the generated image is faithful to the spatial layout of the original input sketch and the image has a natural appearance. Baseline methods, however, exhibit difficulty in maintaining these desired properties in their generations.

this process by iteratively denoising noisy samples, transforming pure noise back into the original data distribution. Several studies, such as DDIM (Song et al., 2021), DPM-solver (Lu et al., 2022), and Progressive Distillation (Salimans & Ho, 2022), have focused on accelerating DMs' generation process through more efficient sampling methodologies. To address the high computational costs of training and sampling, recent research has successfully employed strategies to project the original data into a lower-dimensional manifold, with DMs being trained within this latent space. Representative methods include LSGM (Vahdat et al., 2021), LDM (Rombach et al., 2022), and DALLE-2 (Ramesh et al., 2022), all of which leverage this latent space approach to improve efficiency while maintaining high generation quality.

## 2.2 TEXT-TO-IMAGE DIFFUSION

In addition to producing high-quality and diverse samples, DMs offer superior controllability, especially when guided by textual prompts (Rombach et al., 2022; Xue et al., 2023; Chen et al., 2024; Podell et al., 2024; Esser et al., 2024). Imagen (Saharia et al., 2022) employs a pretrained large language model (e.g., T5 (Raffel et al., 2020)) and a cascade architecture to achieve high-resolution, photorealistic image generation. LDM (Rombach et al., 2022), also known as Stable Diffusion (SD), performs the diffusion process in the latent space with textual information injected into the underlying UNet through a cross-attention mechanism, allowing for reduced computational complexity and improved generation fidelity. To further address challenges when handling complex text prompts with multiple objects and object-attribution bindings, RPG (Yang et al., 2024) proposed a training-free framework that harnesses the chain-of-thought reasoning capabilities of multimodal large language models (LLMs) to enhance the compositionality of T2I generation. Ranni (Feng et al., 2024) tackles this problem by introducing a semantic panel that serves as an intermediary between text prompts and images; an LLM is finetuned to generate semantic panels from text which are then embedded and injected into the DM for direct composition. Our proposed method aligns

with the SD paradigm but diverges by incorporating a composite module that combines textual information with coarse-grained information from sketch inputs, thereby injecting more comprehensive high-level semantics into the diffusion model.

## 2.3 CONDITIONAL DIFFUSION WITH SEMANTIC MAPS

As textual prompts often lack the ability to convey detailed information, recent research has explored conditioning DMs on more complex or fine-grained semantic maps, such as sketches, depth maps, normal maps, etc. Works such as T2I-Adapter (Mou et al., 2024), ControlNet (Zhang et al., 2023), and SCEdit (Jiang et al., 2024), leverage pretrained T2I models but employ different mechanisms to interpret and integrate these detailed conditions into the diffusion process. UniControl (Qin et al., 2023) proposes a task-aware module to unify $N$ different conditions (i.e. $N = 9$) in a single network, achieving promising multi-condition generation with significantly fewer model parameters compared to a multi-ControlNet approach. While Koley et al. (2024) attempts to democratize sketch-based diffusion models, their approach faces several significant limitations, as discussed in the Introduction section. In contrast, our dual-pathway method integrates both fine-grained and coarse-grained sketch conditions while maintaining the option for textual prompts. This design offers greater flexibility and control, accommodating users ranging from amateurs to professionals.

## 3 METHOD

### 3.1 PRELIMINARY

**Stable Diffusion** Diffusion models (Ho et al., 2020) define a generative process by gradually adding noise to input data $z_0$ through a Markovian forward diffusion process $q(z_t|z_0)$. At each timestep $t$, noise is introduced into the data as follows:

$$z_t = \sqrt{\bar{\alpha}_t} z_0 + \sqrt{1 - \bar{\alpha}_t} \epsilon, \quad \epsilon \sim \mathcal{N}(\mathbf{0}, \mathbf{I}), \tag{1}$$

where $\epsilon$ is sampled from a standard Gaussian distribution, and $\bar{\alpha}_t = \prod_{s=0}^{t} \alpha_s$, with $\alpha_t = 1 - \beta_t$ representing a differentiable function of the timestep $t$. The diffusion process gradually converts $z_0$ into pure Gaussian noise $z_T$ over time.

The training objective for diffusion models is to learn a denoising network $\epsilon_\theta$ that predicts the added noise $\epsilon$ at each timestep $t$. The loss function, commonly referred to as the denoising score matching objective, is expressed as:

$$\mathcal{L}(\epsilon_\theta) = \sum_{t=1}^{T} \mathbb{E}_{z_0 \sim q(z_0), \epsilon \sim \mathcal{N}(\mathbf{0}, \mathbf{I})} \left[ \| \epsilon_\theta(\sqrt{\bar{\alpha}_t} z_0 + \sqrt{1 - \bar{\alpha}_t} \epsilon) - \epsilon \|_2^2 \right]. \tag{2}$$

In controllable generation tasks (Zhang et al., 2023; Mou et al., 2024), where both image condition $c_v$ and text prompt $c_t$ are provided, the diffusion loss function can be extended to include these conditioning inputs. The loss at timestep $t$ is modified as:

$$\mathcal{L}_{\text{train}} = \mathbb{E}_{z_0, t, c_t, c_v, \epsilon \sim \mathcal{N}(0,1)} \left[ \| \epsilon_\theta(z_t, t, c_t, c_v) - \epsilon \|_2^2 \right], \tag{3}$$

where $c_v$ and $c_t$ represent the visual and textual conditioning inputs, respectively.

During inference, given an initial noise vector $z_T \sim \mathcal{N}(\mathbf{0}, \mathbf{I})$, the final image $x_0$ is recovered through a step-by-step denoising process (Ho et al., 2020), where the denoised estimate at each step $t$ is calculated as:

$$z_{t-1} = \frac{1}{\sqrt{\alpha_t}} \left( z_t - \frac{1 - \alpha_t}{\sqrt{1 - \bar{\alpha}_t}} \epsilon_\theta(z_t, t, c_t, c_v) \right) + \sigma_t \epsilon, \tag{4}$$

with $\epsilon_\theta$ being the noise predicted by the U-Net (Ronneberger et al., 2015) at timestep $t$, and $\sigma_t = \frac{1-\bar{\alpha}_{t-1}}{1-\bar{\alpha}_t}\beta_t$ representing the variance of the posterior Gaussian distribution $p_\theta(z_0)$. This iterative process gradually refines $z_t$ until it converges to the denoised image $z_0$.

## 3.2 DUAL PATHWAY

Our model introduces a dual-pathway framework that harmonizes high-level semantic abstraction with precise, low-level control over visual details, Figure 5. The integration of the **CGC** module and **FGC** module enables KnobGen to adaptively inject high-level semantics and low-level features throughout the denoising process. This design ensures that the model can scale its output complexity based on user input, thus supporting a wide spectrum of sketch sophistication levels, from amateur to professional-grade sketches.

### 3.2.1 MACRO PATHWAY

Diffusion models typically rely on text-based conditioning using CLIP text encoders (Radford et al., 2021) to capture high-level semantics (Ramesh et al., 2021; Saharia et al., 2022; Nichol et al., 2021), but this approach often misses out on structural cues inherent to other modalities, such as sketches. Although models such as CLIP (Radford et al., 2021) encode visual features and textual semantics, they remain biased toward coarse-grained features (Bianchi et al., 2024; Wang et al., 2023). In our **CGC** module, Figure 5.B, we used this fact to our advantage to fuse a high-level visual and linguistic understanding to control DM generation by incorporating both text and image embeddings through a cross-attention mechanisms.

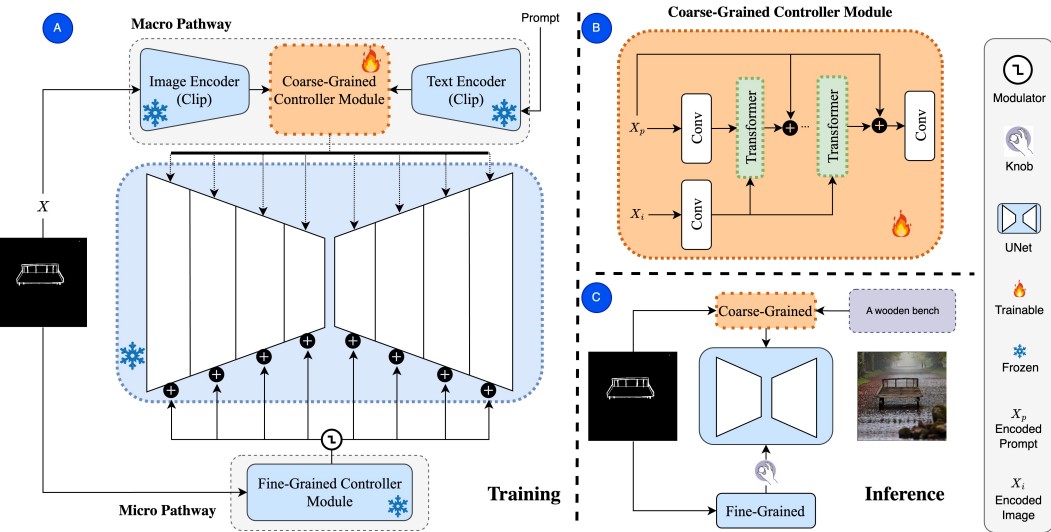

Figure 5: **Overview of KnobGen during training and inference**. A illustrates the training process, where the CGC and FGC modules are dynamically balanced by the modulator. B expands on the CGC module, detailing how high-level semantics from both text and image inputs are integrated. C shows the inference process, including the knob mechanism that allows user-driven control over the level of fine-grained detail in the final image.

**Coarse-grained Controller (CGC):** In our CGC module, we leverage the trained CLIP text encoder and its corresponding image encoder variant available in the pretrained Stable Diffusion Model (Rombach et al., 2022). Our CGC module first takes the raw sketch image (condition) and prompt as input. Using the CLIP image and text encoders, the CGC module first projects them into $x_i \in \mathbb{R}^{256\times1024}$ and $x_p \in \mathbb{R}^{77\times768}$ which are the image and text embeddings. A cross-attention mechanism then fuses these embeddings to produce a multimodal representation that combines textual semantics and visual cues, Figure 5.B. This enables the diffusion process to encode the high-level semantics from text while explicitly integrating spatial features from the sketch using the Clip image encoder. The cross-attended embeddings are injected into layers of the denoising U-Net to

preserve the coarse-grained visual-textual features throughout the diffusion process. Detailed discussion of the CGC module can be found in the Appendix B.

### 3.2.2 Micro Pathway

For artistic users, preserving fine-grained details such as object boundaries and textures is essential. The **Fine-Grained Controller (FGC)** is designed to address these requirements by integrating pretrained modules such as ControlNet (Zhang et al., 2023) and the T2I-Adapter (Mou et al., 2024), which excel in capturing these intricate features. Our Micro Pathway can utilize any pretrained fine-grained controller module which shows the flexibility of our proposed framework.

Incorporating these modules into our micro pathway allows the model to capture detailed, sketch-based features at multiple denoising stages. This pathway complements the coarse-grained features extracted by the CGC module, ensuring that the model not only preserves high-level semantic coherence, but also maintains visual fidelity and spatial accuracy with respect to sketch. Additionally, the FGC module ensures that the model handles professional-grade sketches with precision.

### 3.3 Modulator at training

One of the key innovations in KnobGen is the *tanh-based* modulator, which regulates the contributions of the micro and macro pathways during training, Fig 5.A. Based on our experiments in section 4.4, the incorporation of micro pathway in the early epochs of training process overshadows the effect of our macro pathway. Not only does this phenomenon lead to a model that overfits low-level features of the sketch, but it also prevents the model from generalizing to broader spatial and conceptual features. To mitigate this, we employ a modulator that progressively increases the impact of the Micro Pathway, i.e. the FGC module, during training. The modulator is based on a smooth tanh function:

$$m_t = m_{\min} + \frac{1}{2} \left( 1 + \tanh(\underbrace{k \cdot \frac{t}{T} - 3}_{\psi}) \right) \cdot (m_{\max} - m_{\min}) \tag{5}$$

Here, $t$ is the current epoch, $T$ is the total number of epochs of training, $k = 6$, $\psi \in [-3, 3]$, $m_{\min} = 0.2$ and $m_{\max} = 1$ where $m_{\min}$ and $m_{\max}$ define the range within which the modulator effect (in percent), i.e. $m_t$, will vary over the course of the epochs. In order to choose $m_{\min}$, we heuristically found that the maximum lower bound for negligible effect of the FGC is at $m_{\min} = 0.2$. We did not conduct an extensive hyperparameter search for $m_{\min}$ and only chose this value based on our observation of different case studies, Figure 2. As seen in Figure 5.A, the *module* ensures that diffusion is more affected by the Macro Pathway and less by the Micro Pathway in the early stages of training. As the training progresses, $m_t$ for the Micro Pathway approaches 1 and as a result our FGC module will have an equal impact in the training as that of the CGC. By gradually modulating the influence of the Micro Pathway, we prevent the premature weakening of high-level spatial layout presented by the Macro Pathway, and ensure that both pathways contribute optimally throughout the training process. The effectiveness of our modulator is experimented in section 4.4.

### 3.4 Inference Knob

In typical diffusion models, the early denoising steps during inference focus on generating high-level spatial features, while the later steps refine finer details (Ho et al., 2020; Meng et al., 2021). In our dual-pathway model, this mechanism is explicitly implemented by our proposed *inference-time Knob*. This is essentially a user-controlled tool that determines the range of how much abstraction or rigid alignment with respect to the input sketch is desired by the user, Fig 5.C.

We introduce $\gamma$ variable as our **Knob** parameter. Let the total number of denoising steps be $S$, and $\gamma$ represent the step at which fine-grained details cease to influence the denoising process. The inference knob influence the impact of the CGC and FGC modules at inference-time, allowing users to adjust $\gamma$ depending on their desired level of detail:

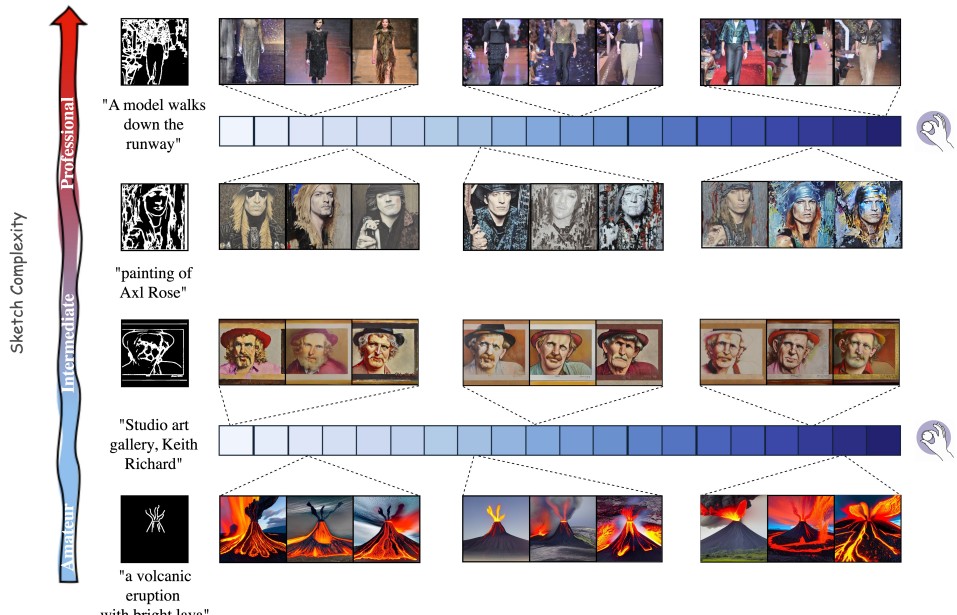

Figure 6: **Impact of the knob mechanism across varying sketch complexities**. From top to bottom, the sketches increase in complexity. The horizontal color spectrum represents the knob values, with light blue on the left ($\gamma$=20) indicating minimal reliance on the sketch, and dark blue on the right ($\gamma$=50) representing maximal reliance.

$$f_\ell(t) = \begin{cases} f_{\text{coarse}}(t) + f_{\text{fine}}(t), & \text{if } t \leq \gamma, \\ f_{\text{coarse}}(t), & \text{if } t > \gamma, \end{cases} \quad \forall \ell \in \{\textit{U-Net layers}\},$$

In this equation, $t$ represents the current denoising step during the inference. The parameter $\gamma$ acts as the knob value, determining the threshold at which the injection of fine-grained features ceases. When the denoising step $t$ is less than or equal to $\gamma$, both coarse-grained features $f_{\text{coarse}}(t)$ and fine-grained features $f_{\text{fine}}(t)$, generated by the macro and micro pathways respectively, are injected into the U-Net across layers, denoted by $\ell$. However, when $t$ exceeds $\gamma$, only the coarse-grained features $f_{\text{coarse}}(t)$ are injected into the U-Net.

A lower $\gamma$ value results in more abstract outputs with respect to the original input sketch, while a higher value makes the model produce images that closely match the sketch's finer details. This adaptive control allows KnobGen to accommodate a wide range of user preferences and input complexities, ensuring that both novice and artists can generate images that align with their expectations, Figure 6. The effectiveness of our proposed Knob mechanism is illustrated in Appendix( C.2).

## 4 EXPERIMENT

We conducted several qualitative and quantitative experiments to validate the effectiveness of Knob-Gen. The qualitative experiments showcase the effectiveness of our approach in guiding the DM based across different sketch complexities. The qualitative experiments evaluate our model against widely-used baselines on different generation metrics such as CLIP and FID scores. We used pre-trained ControlNet and T2I-Adapter as our FGC module throught all our experimentation. According to the parameters defined in section 3.4, $\gamma = 20$ and $S = 50$. These values were heuristically selected and were used consistently in all experiments and baselines.

The extension of the qualitative experiments is available in the Appendix (C). Furthermore, details about the setup used in the training and evaluation are discussed at length in the Appendix (A).

## 4.1 QUALITATIVE RESULTS

Our qualitative results demonstrate the flexibility and effectiveness of KnobGen in handling varying sketch qualities. As shown in Figure 1, KnobGen is able to seamlessly adapt to sketches from rough amateur drawings to refined professional ones, highlighting its ability to cover the entire spectrum of user expertise. Figure 6 illustrates the impact of our knob mechanism, where increasing the knob value (left to right) progressively improves the fidelity to the sketch input. This dynamic adjustment enables precise control over the level of detail, allowing users to fine-tune generation outputs. More qualitative results are provided in the Appendix (C.2).

## 4.2 COMPARISON VS. BASELINES

In order to conduct a fair comparative study, we evaluated KnobGen against baselines such as (Mou et al., 2024; Li et al., 2024; Zhang et al., 2023) on professional-grade sketches, novice ones and a spectrum in between. Figure 4 illustrates the superior quality of the novice-based sketch conditioning using our method against all the other baselines. KnobGen not only captures the spatial layout of the input sketch thanks to the CGC module but also extends beyond it by generating fine-grained details through the FGC module which ultimately produces a naturally appealing images. Whereas the baselines either rigidly conditions themselves on the imperfect input sketch or does not follow the spatial layout desired by the user.

| Models | CNet | T2I | UC | CNet++ | ADiff | KG-CN | KG-T2I |
|---|---|---|---|---|---|---|---|
| CLIP ↑ | 0.3214 | 0.3152 | 0.3210 | 0.3204 | 0.2988 | **0.3353** | 0.3271 |
| FID ↓ | 106.25 | 109.75 | 95.30 | 99.51 | 119.01 | **93.87** | 98.41 |
| Aesthetic ↑ | 0.5182 | 0.5093 | 0.5133 | 0.5253 | 0.4751 | **0.5349** | 0.5208 |

Table 1: Comparison of various models on CLIP score (higher is better), FID score (lower is better), and Aesthetic score (higher is better). The models include ControlNet (CNet), T2I-Adapter (T2I), UniControl (UC), ControlNet++ (CNet++), AnimateDiff (ADiff), KnobGen with ControlNet as the Fine-Grained Controller (KG-CN), and KnobGen with T2I-Adapter as the Fine-Grained Controller (KG-T2I). KnobGen variants (KG-CN and KG-T2I) consistently outperform other models. The number of sketches used for the evaluation is 600.

## 4.3 QUANTITATIVE RESULTS

Table 1 provides a quantitative comparison between state-of-the-art DM models and KnobGen. We evaluated our model with two different FGC module plugins, that is, ControlNet and T2I-Adapter. We call our KnobGen whose FGC module is ControlNet KG-CN and with the T2I-Adapter KG-T2I. We measure performance using the CLIP score (prompt-image alignment), Fréchet Inception Distance (FID) and Aesthetic score (for more information, see Appendix A). KG-CN achieves the highest CLIP score of 0.3353, surpassing the best baseline of 0.3214. KG-CN also gives the lowest FID score (93.87) and the highest aesthetic score (0.5349), demonstrating superior image quality and realism. We use a stratified sampling method based on pixel count to evaluate professional and amateur sketches, ensuring robustness across varying complexity levels. Our results demonstrate KnobGen's effectiveness in generating high-quality images, regardless of input skill level.

## 4.4 ABLATION STUDY

One of the key innovations in our methodology is the introduction of the Modulator, a mechanism designed to enhance the training process of our proposed CGC module. We conducted an experiment where we trained two versions of KnobGen with Modulator and without it. To assess the effectiveness of the Modulator at the inference, we excluded the FGC module after 20 denoising steps in the image generation process ($S = 50$, and $\gamma = 20$, please refer to section 3.4). Excluding the FGC module imposes the conditioning of DM to be done by the CGC module. This experimental configuration demonstrates the power of our CGC module.

Figure 7. presents the results of these experiments, showcasing images generated with and without the Modulator. The comparative analysis reveals that the model trained with the Modulator exhibits

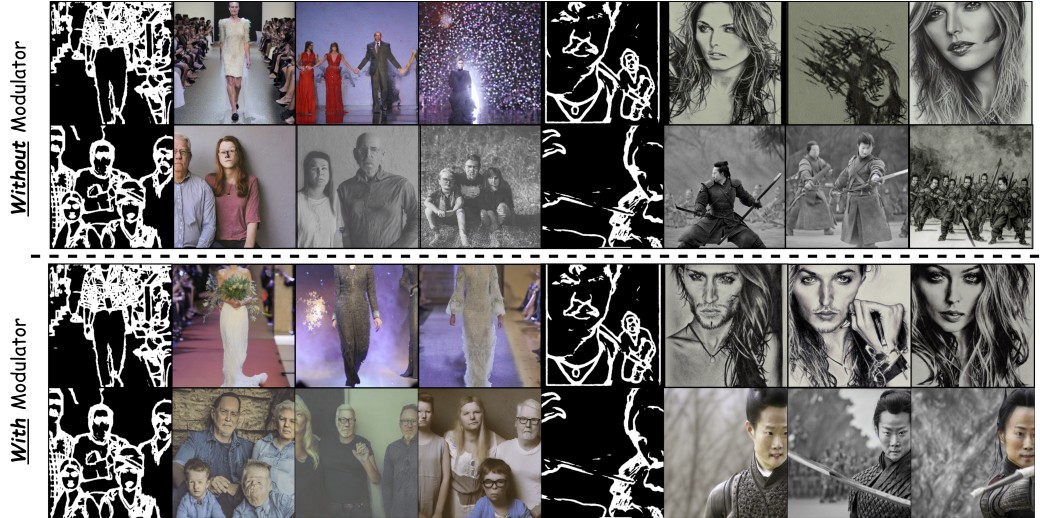

Figure 7: **Comparative results showcasing the impact of the Modulator in the training process**. The top side of the figure displays results generated by the model trained without the Modulator, while the bottom part illustrates outputs from the model trained with the Modulator.

a significantly enhanced ability to integrate *sketch-based coarse-grained guidance* into the image generation process. This indicates that the Modulator not only improves the model's overall performance but also ensures that the CGC's influence is effectively optimized during training, resulting in higher-quality, more accurate image synthesis.

## 5 CONCLUSION

In this paper, we presented KnobGen, a dual-pathway framework designed to address the limitations of existing sketch-based diffusion models by providing flexible control over both fine-grained and coarse-grained features. Unlike previous methods that focus on detailed precision or broad abstraction, KnobGen leverages both pathways to achieve a balanced integration of high-level semantic understanding and low-level visual details. Our novel modulator dynamically governs the interaction between these pathways during training, preventing over-reliance on fine-grained information and ensuring that coarse-grained features are well-established. Additionally, our inference knob mechanism offers user-friendly control over the level of professionalism in the final generated image, allowing the model to adapt to a spectrum of sketching abilities—from amateur to professional. By incorporating these mechanisms, KnobGen effectively bridges the gap between user's input and model robustness. Our approach sets a new standard for sketch-based image generation, balancing precision and abstraction in a unified, adaptable framework.

### REPRODUCIBILITY STATEMENT

Our experiment setups are concisely described in Section 4, with additional implementation details provided in Appendix Sections A and B. We will make our code publicly available to facilitate the reproduction of our results upon acceptance of this paper.

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

# APPENDIX

In this appendix, we provide additional details about the model architecture and supplementary results that further demonstrate the robustness of our approach. These sections aim to provide a deeper understanding of the technical components and showcase more comprehensive comparisons.

- Appendix A: details the setup of training and evaluation in KnobGen.
- Appendix B: expands on the details of the proposed CGC module.
- Appendix C: provides more qualitative results.

## A  SETUP

**Dataset:**  We utilized the MultiGen-20M dataset, as introduced by Qin et al. (2023), to train and evaluate our model. The dataset offers various conditions, making it a suitable choice for our approach. We selected 20,000 images for training, focusing specifically on those with the Holistically-nested Edge Detection (HED) (Xie & Tu, 2015) condition. However, we modified the KnobGen condition by applying a thresholding technique, where pixels below a threshold value of 50 were set to zero, and those above were set to one. This threshold value was chosen through simple visual comparisons of several samples using different thresholds, allowing us to identify the most effective value. This modification essentially transforms the HED condition into a sketch. For evaluation, we curated two distinct sets of images. The first evaluation set consisted of 500 randomly selected samples, which are similar to a sketch drawn by a seasoned artist (we followed the thresholding technique for this part), allowing us to measure our model's effectiveness in professional settings. To further test the robustness and adaptability of our approach, we compiled a second evaluation set of 100 hand-drawn images created by non-professional individuals. This diverse testing set enabled us to demonstrate the model's ability to generalize across a broad spectrum of users, ensuring it can handle both professionally designed and amateur drawings with high robustness.

**Baselines:**  In this work, we evaluate the performance of our proposed model against several state-of-the-art (SOTA) diffusion-based models. Specifically, we conduct both qualitative and quantitative comparisons with prominent models such as ControlNet (Zhang et al., 2023), T2I-Adapter (Mou et al., 2024), AnimateDiff (Guo et al., 2023), UniControl (Qin et al., 2023), and ControlNet++ (Li et al., 2024). These models have achieved significant advances in fine-grained control of image generation by incorporating sketch-based conditions into the diffusion process. Since AnimateDiff is a video-based DM, we only use the first frame of the generated video by it as the comparison point.

**Evaluation:**  We perform qualitative and quantitative evaluation. In the qualitative evaluation, we compare our model's performance across different scenarios of varying input conditions and complexities. For quantitative evaluation, we utilize several metrics to assess the quality of the generated images. First, we calculate the Fréchet Inception Distance (FID) (Heusel et al., 2017; Karras et al., 2019), which measures the similarity between generated and natural images using a pre-trained InceptionV3 model (Szegedy et al., 2016). Lower FID values indicate better generation quality; we used Hukkelås (2020) implementation for our evaluation, which used the default pre-trained InceptionV3 model available in Pytorch (Paszke et al., 2017). To evaluate the alignment between the generated images and the text prompts, we use CLIP (Radford et al., 2021), specifically the pre-trained DetailCLIP model (Monsefi et al., 2024) with a Vision Transformer (ViT-B/16) backbone. Higher CLIP scores signify better alignment between the generated images and their corresponding prompts. Finally, we assess the realism and aesthetic quality of the generated images using the metric proposed by (Ke et al., 2023), where higher scores reflect more realistic and visually appealing images.

**Implementation Details:**  Our proposed *KnobGen* framework is built on top of Stable Diffusion v1.5 (Rombach et al., 2022), with the original parameters kept frozen throughout training. For the Fine-Grained Controller (FGC) module, we employed two different pre-trained models to demonstrate the flexibility and effectiveness of our approach across multiple setups. Specifically, we integrated ControlNet (Zhang et al., 2023) and T2I-Adapter (Mou et al., 2024), both of which had

their parameters frozen and were not updated during training. The architecture and integration of these components are illustrated in Figure 5. We trained the CGC module for a total of 2000 epochs using 16 A100 GPUs. During the initial $1500$ epochs, we employed the modulator mechanism, as described in Section 3.3, with a learning rate of $1e-5$. In the final $500$ epochs, we fine-tuned the CGC model with a reduced learning rate of $1e-6$ to ensure robustness and to improve the quality of the generated images.

## B    MODEL ARCHITECTURE

The CFC module plays a critical role in our model by integrating and aligning visual and textual information for effective image generation. The primary goal of the CFC is to ensure that features derived from both the input sketch image and the text prompt are jointly fused, allowing the model to generate more contextually relevant and visually coherent outputs. The CFC module has around 100M trainable parameters.

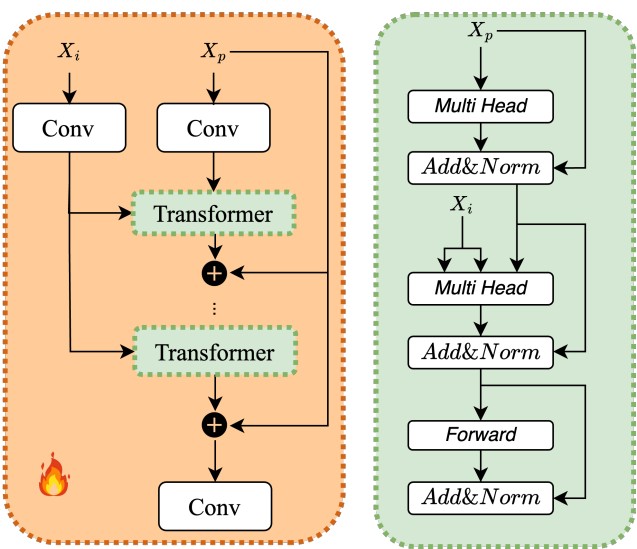

Figure 8: **Overview of the Cross-Feature Conditioning (CFC) module**. The module integrates visual and textual features through a series of transformer blocks with cross-attention. In the diagram, $X_i$ represents the encoded image features from a sketch, while $X_p$ denotes the encoded text prompt. The CFC module conditions the text features based on the image input, allowing for fine-grained control and alignment between visual and textual inputs during the image generation process.

To achieve this, we designed the CFC module using a transformer-based architecture that leverages cross-attention between image and text features; Figure 8 shows the CFC overview. Below, we explain the architecture and functionality in detail:

**Architecture:**   The CFC module is composed of three key components: convolutional layers for feature transformation, transformer layers for cross-attention, and fully connected layers for output projection. The module takes two inputs—visual features (encoded input image) and text features (encoded text prompt)—and processes them jointly to output contextually conditioned text features.

- 1D Convolutional Layers: The input to the CFC module consists of two tensors: an encoded image tensor $x_i \in \mathbb{R}^{\text{batch}\times 256 \times 1024}$, which comes from CLIP image encoder, and an encoded text tensor $x_p \in \mathbb{R}^{\text{batch}\times 77 \times 768}$, which comes from text encoder of CLIP like all the prompt conditioned DM. We then pass these embeddings through 1D convolutional layers to project the input channels (1024 for images and 768 for text) into a common hidden dimension of 1024 channels. This transformation ensures that both modalities can be effectively combined in the cross-attention mechanism.
- Transformer Layers for Cross-Attention: The core of the CFC module lies in its eight layers of transformers that perform cross-attention. These layers allow the model to fuse infor-

mation from both the image and text features. Specifically, the image tensor serves as the memory input for the transformer, while the text tensor undergoes cross-attention, attending to the visual information. This design enables the model to enhance text-based features by conditioning them on the spatial and structural content of the image. The resulting enriched text features better capture the contextual relevance of the image, leading to more semantically meaningful generation.

- Fully Connected Layers: After passing through the transformer layers, the output text tensor is reduced back to its original sequence length (77 tokens) and further processed through two fully connected layers. These layers refine the text features, ensuring that the final output has the desired dimensionality (batch, 77, 768) and captures the relevant information for conditioning the image generation process.

**Reasoning Behind the Design:**   The CFC module is specifically designed to address the need for strong alignment between visual and textual inputs during image generation. By using a cross-attention mechanism, the module ensures that the text features are not treated independently of the visual content, but rather, are conditioned on the image's features as well. This approach is particularly useful when fine-grained control is needed to generate images that aligns to both the textual description and visual input, making it highly effective in scenarios where accurate text-to-image alignment is crucial. Additionally, the use of pre-trained models ensures that the model benefits from robust initial feature extraction which further improves generation quality as a result.

## C   MORE QUALITATIVE RESULT

This section contains more qualitative results to complement the evaluations presented in the main paper. We provide visual examples of different use cases, including scenarios involving amateur and professional sketches.

### C.1   INFERENCE KNOB MECHANISM FOR BASELINES

One of the important ablation studies was to evaluate the performance of fine-grained controller models, such as the T2I-Adapter, when they utilize our Knob mechanism. This ablation study was particularly performed to demonstrate the effectiveness of our proposed CGC module.

Models such as T2I-Adapter are traditionally designed for precise, detail-oriented image generation but lack the flexibility to accommodate broader, more abstract inputs like rough sketches or varying user skills. To explore this issue, we integrated the Knob system into the T2I-Adapter model **without our CGC module**.

Figure 9 showed that while the T2I-Adapter performs exceptionally well in generating high-fidelity images from professional-grade inputs, it struggles to maintain this quality when dealing with rougher or less detailed sketches. This limitation arises from the absence of a Macro Pathway in the T2I-Adapter's architecture, which makes the model overly reliant on precise input details. Without the ability to capture broader, high-level semantic information through a coarse-grained approach, the model becomes highly sensitive to adjustments made by the Knob mechanism. As a result, T2I-Adapter fails to deliver consistently good results across a diverse range of users, particularly those providing amateur or less-defined sketches. Additionally, we observed that after a certain point, increasing the Knob value no longer meaningfully affects the generation output. This suggests that the sketch condition in T2I-Adapter influences the generation primarily in the early denoising steps, with diminishing effects in the later steps. However, further investigation of this behavior is outside the scope of this study.

While the Knob system is designed to balance coarse and fine-grained controls dynamically, the lack of a dedicated coarse-grained module in T2I-Adapter causes the model to lose spatial coherence when we apply our Knob mechanism for it, especially when the knob has low value. This issue became particularly evident when trying to generate images based on prompt only, as the model struggled to infer the missing spatial structure, leading to distorted or incoherent outputs.

In contrast, the KnobGen framework, including the CGC and FGC, demonstrated superior flexibility and performance. By incorporating both high-level abstractions and detailed refinements, KnobGen

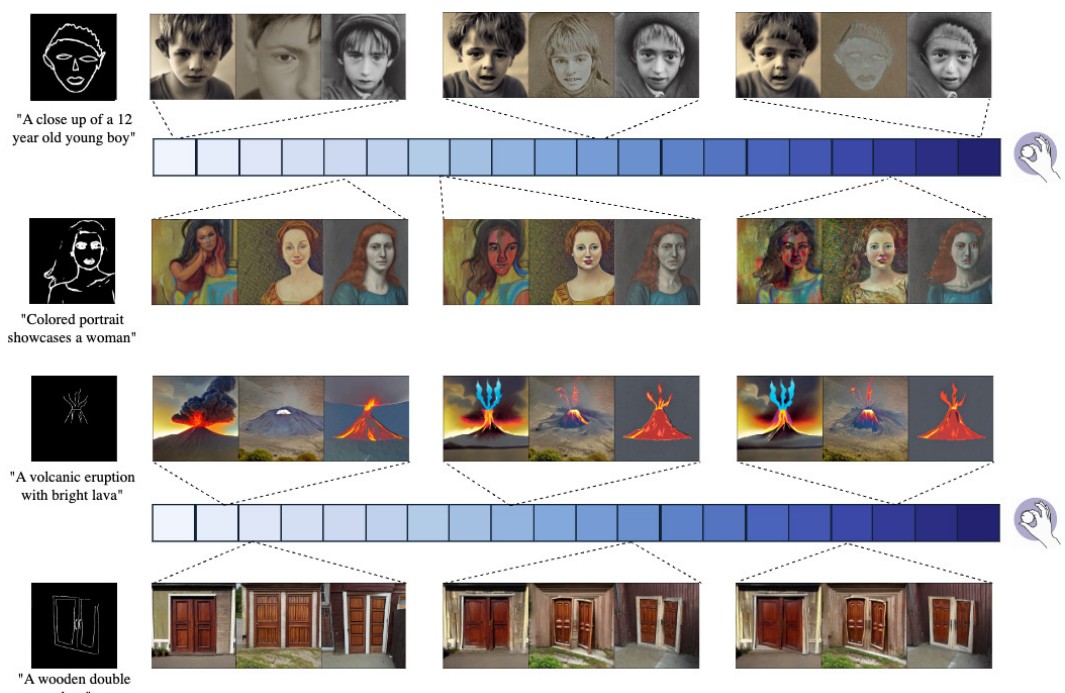

Figure 9: **Effect of the Knob mechanism on the fine-grained models (T2I-Adapter)**. The image demonstrates how increasing the Knob value influences the generated output. While the T2I-Adapter performs well with precise, detailed sketches, it struggles with rougher sketches and fails to maintain spatial consistency as the Knob value increases. Beyond a certain threshold, the sketch has minimal impact on the final output, highlighting the model's sensitivity to early-stage adjustments and its limitations in handling coarse-grained information.

could adapt dynamically to the varying levels of detail in the input sketches. The CGC in KnobGen helps preserve the overall structure and semantics of the image, while the FGC ensures that fine details are accurately rendered.

## C.2 MORE QUALITATIVE RESULTS

In this section, we present additional qualitative results to demonstrate the effectiveness and versatility of our proposed KnobGen framework further. Figure 10 and 11 showcases the model's ability to handle a wide range of input sketches, from highly detailed professional-grade drawings to rough, amateur sketches.

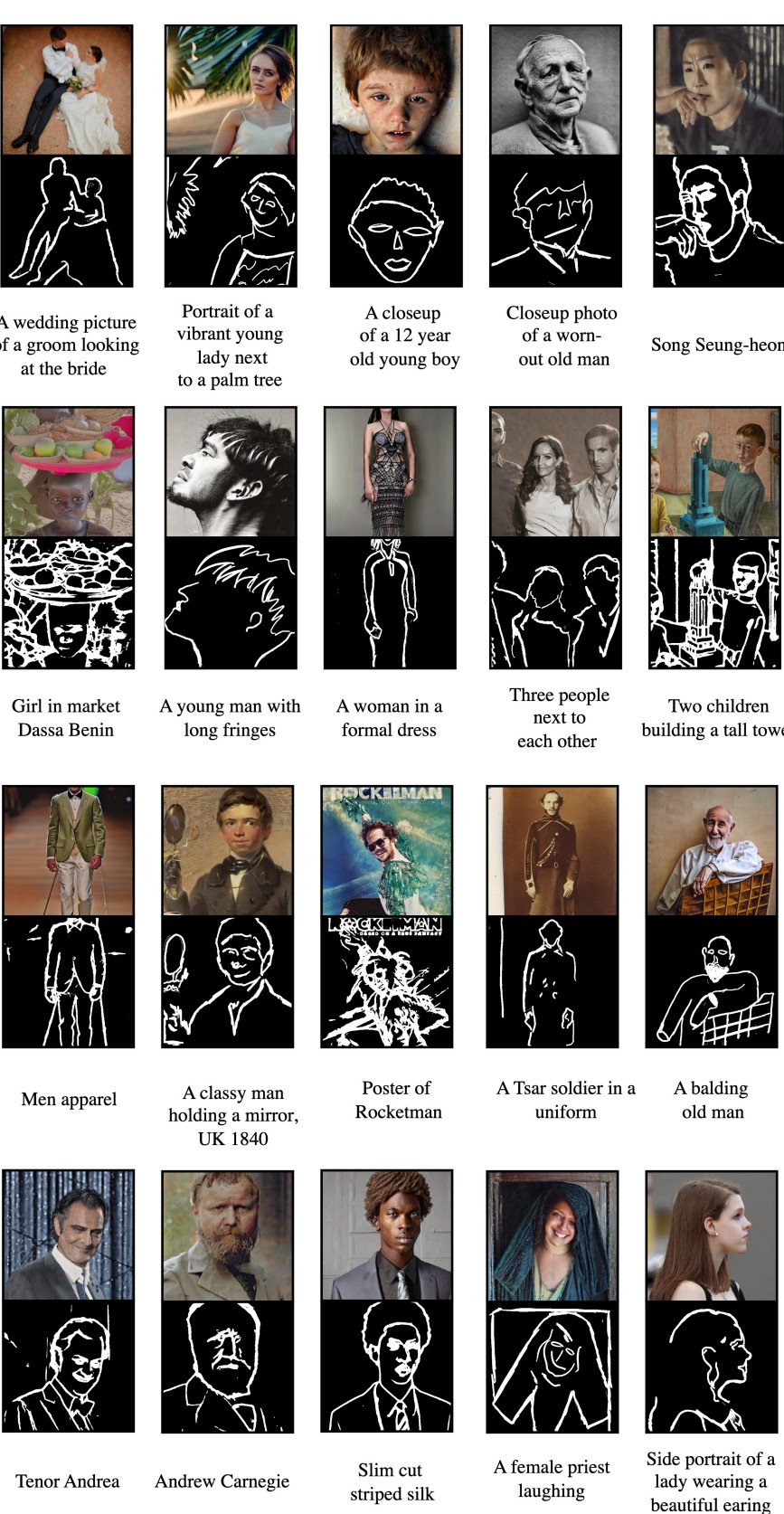

Figure 10: More qualitative results on novice and professional-grade sketches

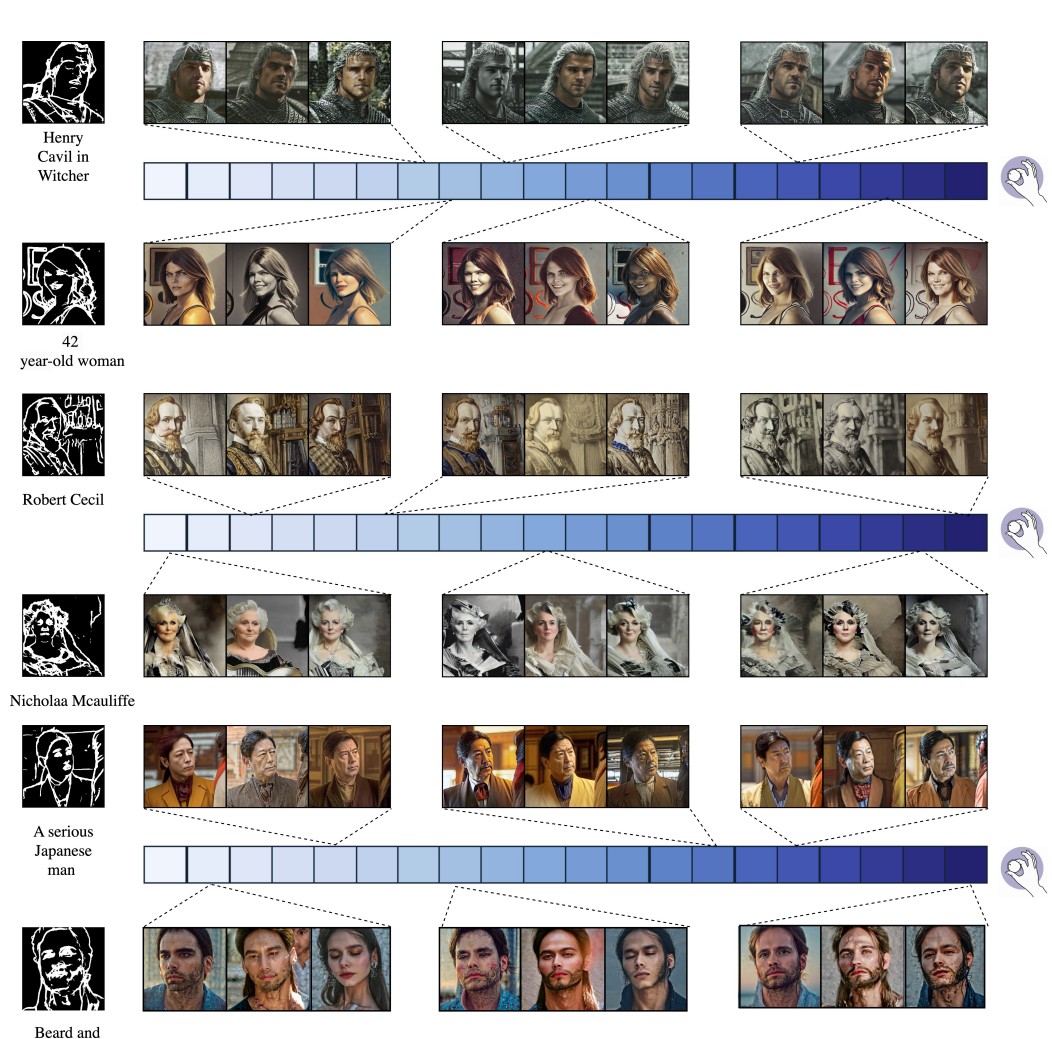

Figure 11: More knob spectrum results

