# OpenReview forum: "KnobGen: Controlling the Sophistication of Artwork in Sketch-Based Diffusion Models"
_ICLR.cc/2025/Conference — ICLR 2025 Conference Withdrawn Submission_

### Official Review · Reviewer_rE5D · 2024-10-24

**Soundness:** 2
**Presentation:** 3
**Contribution:** 2
**Rating:** 5
**Confidence:** 3

**Summary:**

In this paper, the authors propose a sketch-based image generation method which can generate high-quality images with varying levels of sketch complexity, named KnobGen. It designs a coarse-grained controller module and a fine-grained controller module to learn high-level semantics and detailed refinement separately. Then the authors introduce a knob inference mechanism to align with the user’s specific needs.  The experiments have shown superior performance compared to other sketch-based images generation methods.

**Strengths:**

1.	KnobGen introduces a dual-pathway framework that balances the effects of both coarse-grained and fine-grained features, resulting in better performance compared to other sketch-based image generation methods.

2.	The authors introduce a dynamic modulator to regulate the influence of the CGC and FGC modules during training, thereby preventing premature overfitting to fine details.

**Weaknesses:**

1. Although the proposed method balances the effects of coarse-grained and fine-grained features, it may not meet users' expectations when the input sketches are highly abstract. For example, as depicted in the last column of the second and third rows in Figure 4, the generated images align with the distortions shown in the sketches. Although this effect can be adjusted by the hyperparameter γ, changing the “controllable scale” within the ControlNet and T2I-adapter can also yield similar outcomes.  To be specific, the authors are recommended to provide additional results based on varying the 'controllable scale' of ControlNet and T2I-adapter. This would illustrate that the proposed method can meet user needs without the need for adjusting γ.

2. Setting a hyperparameter γ to control the conditional signal based on the denoising steps during inference is a common operation in T2I task, which is not a novel technical contribution. The authors are suggested to demonstrate the differences between this method and previous ones. For instance, they could illustrate the relationship between the tanh-based modulator used in training and the γutilized during inference.

3. There are still some experimental issues:

(1). The evaluation of the generated results is somewhat subjective. Additional user studies could further demonstrate the strengths of the proposed method. Specifically, the authors can prepare more than 50 sets of sketches, which need to be evenly distributed across different levels of complexity. Users can then rank the results generated by the proposed method and other sketch-based methods.

(2). Since these models that assess user preferences are trained on their own designed datasets, a single aesthetic score is insufficient. The authors are suggested to conduct quantitative experiments using the HPS-v2[A] and Pick score[B] metrics, which are widely used for comprehensively assessing the generation quality.

[A]. Wu X, Hao Y, Sun K, et al. Human preference score v2: A solid benchmark for evaluating human preferences of text-to-image synthesis[J]. arXiv preprint arXiv:2306.09341, 2023.

[B]. Kirstain Y, Polyak A, Singer U, et al. Pick-a-pic: An open dataset of user preferences for text-to-image generation[J]. Advances in Neural Information Processing Systems, 2023, 36: 36652-36663.

(3). In the ablation study, there are too few examples, as the displayed images only include results of human portrait generation. The authors are recommended to add results of sketch generation for other categories.  And the authors are recommended to provide more quantitative ablation results, with the same metrics as the quantitative experiments to demonstrate the effectiveness of the proposed method.

**Questions:**

Please refer to the questions in the “Weaknesses” part.

---

### Official Review · Reviewer_7Msb · 2024-11-03

**Soundness:** 2
**Presentation:** 1
**Contribution:** 2
**Rating:** 1
**Confidence:** 5

**Summary:**

In this paper, the authors introduce an new dual-pathway framework that aims to overcome the shortcomings of existing sketch-based diffusion models. By considering both fine-grained and coarse-grained features, this approach elegantly combines high-level semantic understanding with low-level visual details. The proposed method not only enhances the connection between user input and model robustness but also paves the way for more effective results in sketch-based tasks.

**Strengths:**

1. The authors provide visually appealing figures that make the content engaging and easy to understand.
2. Comprehensive experiments demonstrate the effectiveness of the proposed method, showcasing its potential in real-world applications.

**Weaknesses:**

1, I can not follow this paper. The writing is bad, since there are many sentences I can not understand.  In introduction,  the presentation is not good.  (1)  ‘Furthermore, we observed that the quality and alignment of the generated images with the input sketch are highly sensitive to the weighting parameter that governs the model’s dependence on the condition, Figure 2. (2) 'Additionally, the removal of the text-based conditioning in DM makes these models ignore the semantic power provided by text in diffusion models trained on large-scale image-text pairs, Additionally,...'. There are two ‘Additionally’ , which is not enjoyable.  (3)  'Professional-oriented models like ControlNet and T2I-Adapter are designed to handle only artistic-grade sketches Fig. 3.a, while amateur-oriented approaches Koley et al. (2024), cater to novice sketches without text guidance Fig. 3.b'.  In some places it is Figure , while some ones are Fig..  (4) 'The Macro Pathway extracts the high-level visual and language semantics from the sketch image and the text prompt using CLIP encoders and injects them into the DM via our proposed Coarse-Grained Controller (CGC)'.  too much ‘ and’

2. Some figure captions lack clarity; for instance, in Figure 2, the term “weights” could use more explanation.

3. There are concerns that the overall writing style may resemble that produced by a language model, suggesting a need for a more personal touch.

**Questions:**

The writing is not good, since there are many sentences we can not understand.

---

### Official Review · Reviewer_o6Zw · 2024-11-03

**Soundness:** 2
**Presentation:** 2
**Contribution:** 2
**Rating:** 5
**Confidence:** 5

**Summary:**

This work proposed a sketch-based image generator that accepts sketches with different complexities from novice users and professionals. And reasonable outcomes (i.e., high-quality images) can be obtained in either case. At its centre is the proposed dual-path way conditional diffusion model, which can handle the coarse-grained and fine-grained conditions separately and simultaneously. Experimental results validated the flexibility and effectiveness of the proposed method for sketch-to-image generation.

**Strengths:**

- The proposed method works for input sketches with varying abstraction levels, ranging from amateur and professional sketches. This inclusiveness is meaningful in practice as it can expand the scope of end-users with different needs and drawing skills.
- The design of the modulator during model training is interesting. The goal is to harmonise the dual-path conditions, thus achieving more enjoyable results. The intuition to let the coarse-grained cues dominate at the early denoising steps and gradually increase the impact of fine-grained cues is sound.
- The degree of alignment between the input sketches and the generated images is controllable during the inference. Such flexibility is desirable for boosting the applicability of sketch-guided text-to-image generation.

**Weaknesses:**

- The professional sketches (Multi-Gen-20M) considered in this work are in binarised versions of HED edges, which is very different from what a real artist would draw (no artist or professional sketcher would produce lines like those in Figure 1). This makes the basic assumptions/conditions of the paper not very rigorous, somewhat deviating from the ambitious objectives, i.e., dealing with pro-sketch and any other complexity levels with a unified model.
- The modulator is heuristically designed. It is hard to justify if there is a scalability issue that might need tedious hyperparameter tuning for diverse training data.
- The effectiveness and applicability of the knob mechanism is questionable.
    - From Figure 6, the effect does not seem very pronounced: in the volcano example, the volcano corresponding to the intermediate gamma value appears to match the details of the input sketch better; in Keith's example (the second row from the bottom), the changes in facial details are also not noticeable.
    - Besides, the user has to try different knob values until satisfaction (and this may be pretty different for diverse input sketches) since it has no apparent relation to the user's need for the complexity level from the input sketches.
    - The impact of fine-grained cues is hard to manage precisely, as they have been injected into the model at early denoising steps, and the effect will last in the following denoising steps.
- The current competitors in experiments are not designed for sketches. It would be great if some sketch-guided image generation works, e.g., [a], could be compared and discussed.
- There is a “second evaluation set” with 100 hand-drawn images created by novice users used for experiments. It would be great to show these sketch images for completion.

[a] Sketch-Guided Text-to-Image Diffusion Models, SIGGRAPH 2023

**Questions:**

Please refer to weaknesses.

---

### Official Review · Reviewer_MNa1 · 2024-11-03

**Soundness:** 2
**Presentation:** 2
**Contribution:** 2
**Rating:** 5
**Confidence:** 5

**Summary:**

The paper proposes "KnobGen", a novel algorithm for sketch+text-based image generation and provides the user control over the balance between fine-grained and coarse-grained queries coming from the input sketch and textual prompt respectively. It proposes a dual-pathway framework that democratises sketch+text-based image generation by adapting to varying levels of sketch abstraction and user-skills. It proposes a Coarse-Grained Controller (CGC) block for high-level semantics and a Fine-Grained Controller (FGC) block for further refinement of fine-grained features in the final output. With the possibility of controlling the relative strengths of these modules, this method can control how fine or coarse grained the final output will be. Finally, it also proposes a new sketch dataset on top of the MultiGen-20M dataset. Experimental results shows this method to outperform other popular methods like ControlNet, T2I-Adapter, etc.

**Strengths:**

[+] The paper is easy to read, and generally well written.

[+] The work addresses the research gap of allowing user control the influence of sketch and text on the generated image useful for end-users ranging from novice to professionals.

**Weaknesses:**

[-] The paper includes a qualitative and quantitative comparison with several SoTA methods. However, a thorough quantitative ablation study examining the impact of each component would greatly enhance the reader's understanding of their relative importance. There exists an ablative study but it appears to be superficial and does not justify all design choices made in the paper.

[-] The quantitative results provided in the work lacks critical information regarding the value of knob used -- necessary to gauge the improvement of the proposed model. It would have been great to see the changes quantitatively on how the actual numeric values of the knob affecting the final output metrics.

[-] Generally the contribution factor of both micro and macro pathways should sum to 1. However, in this paper, the knob during training increases the contribution of micro pathway in each epoch, without decreasing the macro pathway. Won't doing this cause issues in the overall magnitude of the feature map (as one pathway is getting more weightage than the other)?

[-] Any particular reasoning behind using a "tanh" instead of any other function in knob during training for increasing the contribution of micro pathway? It would have been great to see ablation in this aspect as well, to fully justify this design choice.

[-] Other SoTAs like ControlNet also provides a balancing factor to control sketch-conformity. The proposed method should be compared in this aspect as well to check how granular the control is compared to SoTAs.

[-] It would be very interesting to see "sketch-only" generation results, either by passing null-prompt or making the text-pathway weighting factor to zero.

**Questions:**

See weaknesses

---

### Note · Authors · 2024-11-14

**Comment:**

Subject: Concerns Regarding Reviewer 7Msb's Conduct and Request to Withdraw Submission

Dear ICLR Program Chair,

I am writing to formally express my concerns about the quality and professionalism of the review provided by Reviewer 7Msb for my paper submitted to ICLR 2025. While constructive feedback is invaluable, one of the review I received that happened to be a "strong reject" decision was strikingly partial and dismissive, focusing almost exclusively on superficial language aspects rather than engaging with the paper’s technical contributions. Given this misalignment, I have decided to withdraw my submission.

My concerns with Reviewer 7Msb’s evaluation are as follows:

1.	The review provides vague and generalized criticisms of the “presentation” and “writing style,” including comments such as “The writing is bad.” This unconstructive critique lacks specificity, offering no actionable insights or suggestions for improvement.

2.	Despite acknowledging the innovation and potential impact of the proposed dual-pathway framework, Reviewer 7Msb inexplicably assigned a low score for “Soundness.” This score was assigned without any technical analysis or critique of the framework, methodology, or experimental results, calling into question the review’s rigor and impartiality.

3.	The reviewer focused disproportionately on minor language and stylistic choices, such as repeated terms or caption wording, without recognizing these elements' supporting roles in the paper's broader technical narrative.

Particularly concerning is Reviewer 7Msb’s speculative remark that the writing “resembles that produced by a language model.” Such a comment is both inappropriate and irrelevant, diverting attention from the paper’s primary technical contributions and raising questions about the reviewer's objectivity.

Given these issues, I feel compelled to withdraw my submission. I hope that by sharing this feedback, I can underscore the importance of upholding high standards of objectivity and professionalism in the ICLR review process. These standards are vital to maintaining the conference’s standing as a leading forum for innovative AI research.

Thank you for your attention to this matter.

Sincerely,
Pouyan B. Navard
PhD Candidate at the Ohio State University

**Withdrawal Confirmation:**

I have read and agree with the venue's withdrawal policy on behalf of myself and my co-authors.